# Semi-Supervised Gastrointestinal Stromal Tumor Detection via Self-Training

**Qi Yang [1,†], Ziran Cao [2,†], Yaling Jiang [3], Hanbo Sun [2], Xiaokang Gu [2], Fei Xie [4,5,*], Fei Miao [6,*] and Gang Gao [7]**

1. Department of Neurology, National Center for Neurological Disorders, Huashan Hospital, Fudan University, Shanghai 200433, China
2. College of Information Science and Technology, Northwest University, Xi'an 710127, China
3. School of Computer Science and Technology, Xidian University, Xi'an 710071, China
4. Frontier Cross Research Institute, Xidian University, Xi'an 710071, China
5. Xi'an Key Laboratory of Human-Machine Integration and Control Technology for Intelligent Rehabilitation, Xijing University, Xi'an 710123, China
6. Department of Ultrasound, Ruijin Hospital, Shanghai Jiao Tong University School of Medicine, Shanghai 200025, China
7. Shanghai Yiran Health Consulting Co., Ltd., Shanghai 201821, China
* Correspondence: fxie@xidian.edu.cn (F.X.); mf11066@rjh.com.cn (F.M.)
† These authors contributed equally to this work.

**Abstract:** The clinical diagnosis of gastrointestinal stromal tumors (GISTs) requires time-consuming tumor localization by physicians, while automated detection of GIST can help physicians develop timely treatment plans. Existing GIST detection methods based on fully supervised deep learning require a large amount of labeled data for the model training, but the acquisition of labeled data is often time-consuming and labor-intensive, hindering the optimization of the model. However, the semi-supervised learning method can perform better than the fully supervised learning method with only a small amount of labeled data because of the full use of unlabeled data, which effectively compensates for the lack of labeled data. Therefore, we propose a semi-supervised gastrointestinal stromal tumor (GIST) detection method based on self-training using the new selection criterion to guarantee the quality of pseudo-labels and adding the pseudo-labeled data to the training set together with the labeled data after linear mixing. In addition, we introduce the improved Faster RCNN with the multiscale module and the feature enhancement module (FEM) for semi-supervised GIST detection. The multiscale module and the FEM can better fit the characteristics of GIST and obtain better detection results. The experiment results showed that our approach achieved the best performance on our GIST image dataset with the joint optimization of the self-training framework, the multiscale module, and the FEM.

**Keywords:** gastrointestinal stromal tumor; semi-supervised learning; self-training; object detection; computational intelligence

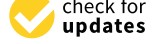



## 1. Introduction

A gastrointestinal stromal tumor (GIST) is a highly aggressive gastrointestinal mesenchymal tumor, mainly diagnosed with image examination. GIST detection on abdominal CT images helps acquire the location of tumors, formulate treatment plans in time, and prevent distant metastasis of tumors. Currently, computer vision technology based on the neural network has been widely used in a variety of lesion detection tasks, including the diagnosis of pulmonary tuberculosis, breast cancer, and other lesions. Unlike lesion detection in other organs, the small intestine has a motor function, so the shape of the GIST varies widely, leading to low discriminability and many difficulties for GIST detection. Therefore, there are few studies related to the detection and identification of GIST, and existing techniques still use common object detection algorithms (Faster R-CNN [1], YOLO [2],

Cascade R-CNN [3], RetinaNet [4], etc.). These fully supervised learning methods heavily rely on labeled data for the parameter optimization, but the medical images must be labeled by doctors with clinical experience, which requires considerable resources. As a result, the detection results of neural networks are affected by the lack of labeled data. Meanwhile, a substantial number of unlabeled images are stored in the hospital's medical system. Compared with the high cost of manually labeling them, it is much easier to retrieve these images. To solve the problem of unsatisfactory training results caused by the imbalance between the number of labeled and unlabeled images, we use the semi-supervised learning method to train the model.

With limited labeled data, the semi-supervised learning (SSL) method can improve the performance of the model by effectively using a tremendous amount of unlabeled data to optimize the model while reducing the dependence on labeled data. Generally, the SSL method can be divided into two steps: (a) training on a small amount of labeled data to obtain model A and predicting pseudo-labels of unlabeled data through A; and (b) retraining the model on a new dataset consisting of pseudo-labeled and labeled data to improve the performance of the model. Since the pseudo-labels of the unlabeled data are generated with the model prediction, there may be some mistakes. Some researchers [5,6] have used self-integration methods to improve the quality of pseudo-labels and enhance the robustness of the model. In addition, there are also algorithms [7,8] that learn complementary information by cotraining to avoid confirmation bias and guarantee the accuracy of pseudo-labels. In order to prevent wrong pseudo-labels from producing errors that continue to iterate and affect the performance of the model, we propose a self-training-based SSL method (Figure 1) that uses the dual constraints of dynamic threshold and IOU to enhance the quality of pseudo-labels. The dynamic threshold constraint means setting a minimum threshold for the confidence of the pseudo-label, using a higher confidence threshold at the beginning of the training and gradually decreasing it as the training progresses. The IOU constraint means that the intersection over union (IOU) between multiple pseudo-labels of different transformed images should be greater than the set threshold; that is, after the data augmentation on the unlabeled image is completed, the shape and the position of the new candidate bounding box should maintain a certain immutability.

Additionally, to fundamentally ensure the quality of pseudo-labels, we improved the popular two-stage object detector (Faster R-CNN [1]) and applied the Improved Faster R-CNN to pseudo-label generation. Most of the traditional object detection algorithms used in existing GIST detection techniques are designed for object detection tasks in natural images and perform well on salient object detection. However, GIST has an unclear boundary and inconspicuous features in abdominal CT, and these algorithms cannot achieve good results. Compared with these techniques, Improved Faster R-CNN makes adjustments targeted to the morphological characteristics of GIST. In Improved Faster R-CNN, the multiscale module and the feature enhancement module (FEM) designed for the characteristics of GIST have been added. The newly added module can better detect GISTs of different scales in complex backgrounds, which helps to improve the accuracy of pseudo-labels. Finally, Mixup [9] can be used to augment the true labels of the labeled data and the pseudo-labels of the unlabeled data. The generalization capability of the network can be significantly enhanced by linearly mixing these samples.

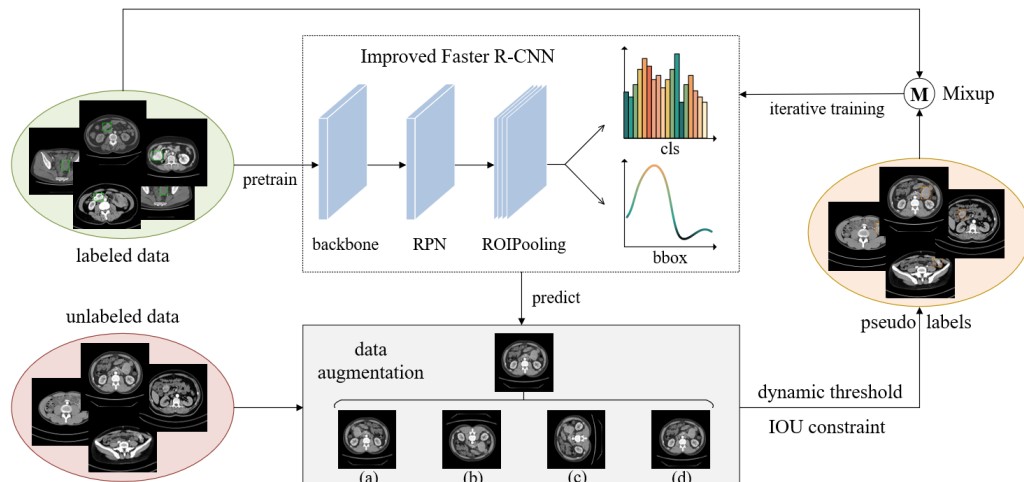

**Figure 1.** Overview of the self-training method. In each iteration, the predictions of the Improved Faster R-CNN on unlabeled data are augmented and then filtered out with the IOU constraint and the dynamic threshold together to generate pseudo-labels. The labeled data and the pseudo-labeled data are linearly mixed using Mixup, and the resulting new dataset is used for the next iteration. The data augmentation methods in the figure are (a) horizontal flip, (b) vertical flip, (c) rotation, and (d) affine transformation; one is randomly selected as the data augmentation method and repeated k times during the experiment.

In summary, our main contributions are as follows: (1) We propose a detection algorithm (Improved Faster R-CNN) for GIST detection and use it as the benchmark model for the SSL method. (2) We propose a novel self-training-based SSL method for GIST detection. (3) Extensive experiments demonstrated that the performance of the proposed SSL method is significantly improved compared to the fully supervised learning method.

## 2. Related Work

### 2.1. Lesion Detection

Lesion detection is an important computer vision task in the field of CAD (computer-aided diagnosis) and has received considerable attention in recent years. Many scholars have designed excellent object detectors based on convolutional neural networks (CNNs) for lesion detection. Cireşan et al. [10] added max-pooling layer and postprocessing strategies to the CNN for mitosis detection on mammary gland histological images. Setio et al. [11] proposed a multiview convolutional network that combines the respective advantages of three detectors for pulmonary nodule detection. Rajpurkar et al. [12] improved the dense convolutional network by replacing the fully connected layer with a single output layer and applying a nonlinear sigmoid activation function to achieve excellent performance on the task of pneumonia detection on chest radiographs. Sedik et al. [13] constructed a deep learning architecture for COVID-19 detection on CT images and X-ray films based on CNN and ConvLSTM, which included convolutional, pooling, and ConvLSTM layers, and the multilayered structure effectively reduced the overfitting errors and enhanced the detection accuracy. Although artificial intelligence technologies have been widely used in the field of CAD, there is still little research on GIST detection. The small intestine moves by its nature, and the GIST appears with considerable morphological differences in abdominal CT, resulting in difficulty in improving the accuracy of GIST detection. At present, only our team has carried out this research. Fei et al. [14] have combined a variety of classical fully supervised detection algorithms to improve the accuracy of GIST detection, but the theoretical innovation of this method needs to be improved. In addition, the method requires training on a large amount of accurate manually annotated data, which is expensive and time-consuming to acquire in the medical imaging domain. In contrast

to previous methods that solely train models on labeled data, our SSL approach trains an object detector on both labeled and unlabeled data.

### 2.2. Semi-Supervised Object Detection

Semi-supervised learning methods can leverage latent knowledge from unlabeled data to facilitate model learning with limited labeled data [15]. Existing SSL methods consist of two categories: consistency-based methods and self-training-based methods. The main idea of the consistency-based [16–18] approach is that for any input data, its output should be consistent with the original output when it is disturbed by less noise. Self-training-based approaches improve the performance of SSL by filtering noisy labels using a predefined threshold and adding the retained pseudo-labels into model retraining. Lee et al. [19] used the deep neural network to train both labeled and unlabeled data simultaneously and pioneered the method of using pseudo-labels for training. Iscen et al. [20] used a transduction label propagation method based on the prevalence hypothesis in predicting pseudo-labels and achieved transduction learning by calculating the similarity matrix between the labeled and unlabeled data. Qizhe Xie et al. [21] improved the quality of pseudo-labels through repeated teacher–student model iterations to enhance the robustness and accuracy of self-training. Considering the uncertainty of the teacher network in the self-training method, Mukherjee et al. [22] chose the Bayesian network to estimate the uncertainty of pseudo-labels, thereby reducing the influence of noisy labels on the model.

The SSL method has also been widely used in the object detection field, and many researchers are committed to training high-performance object detectors with a limited amount of labeled data and a large amount of unlabeled data. Jeong et al. [23] proposed the CSD method based on consistency regularization, which calculates the consistency loss between the prediction on the original unlabeled image and the flipped unlabeled image to achieve the aim of fully utilizing unlabeled data. Sohn et al. [24] combined both self-training and consistency regularization to propose the STAC method, which first eliminates some low-confidence pseudo-labels obtained from self-training by threshold screening and calculates unsupervised loss as well as a supervised loss while training on the augmented unlabeled data together with labeled data. Qize et al. [25] performed self-training object detection based on the mean teacher model, using the nonmaximum suppression (NMS) method to fuse the detection results from different iteration periods to ensure the stability of the detection results during the training process. Moreover, the use of double-head can effectively utilize the complementary information and improve the quality of the pseudo-labels. To address the problem of the imbalance between the foreground and background, Fangyuan et al. [26] proposed a self-training method of adaptive class rebalancing that stores and extracts foreground instances and pastes them into random positions of training samples, increasing the proportion of foreground instances. They also designed a two-stage filter to weed out unreliable pseudo labels.

Although the SSL method for object detection has garnered a degree of success, the following problems in SSL-based GIST detection still remain: (1) The current semi-supervised object detection methods only employ basic object detection algorithms, such as Faster R-CNN [1]. However, given that GIST does not have clear boundary features in the CT image and has a large degree of variation, it is necessary to use an object detection algorithm that is more suitable for these characteristics. (2) The current object detection approach uses self-training without taking into account erroneous pseudo-labels, which results in overfitting to the wrong pseudo-labels and a decrease in the model's accuracy. As a result, we have built our SSL framework using the Improved Faster R-CNN with the multiscale module and the FEM. We then added the dynamic threshold and the IOU constraint in the self-training process to increase the accuracy of pseudo-labels, ensuring that the following model iterations perform better.

## 3. Method

With a severe lack of labeled data, using limited labeled data to improve model performance has become a significant problem in GIST detection. To fully utilize all data, including unlabeled data, we propose an SSL method based on self-training and design Improved Faster R-CNN as the detection algorithm according to the characteristics of GIST. The Improved Faster R-CNN containing the multiscale module and the FEM can better integrate multidimensional feature information and combine deep semantic information with shallow location information. We further developed a new pseudo-label selection strategy to improve the robustness of the model. By applying the dynamic threshold constraint and the IOU constraint to the prediction results of unlabeled data, the reliable pseudo-labels can be retained for subsequent training. In the subsequent sections, we describe improvements to the Faster R-CNN by introducing two new modules aimed at characterizing GISTs (Section 3.1). Next, we introduce a novel pseudo-label selection strategy (Section 3.2) and outline our self-training approach (Section 3.3).

### 3.1. Improved Faster R-CNN

We optimized the Faster R-CNN to improve the accuracy of GIST detection and ensure the quality of pseudo-labels; the network structure is shown in Figure 2. In this paper, two optimization modules are proposed: (1) Given the large variability of GISTs, a multiscale module was developed to use feature information of various levels. (2) The FEM was introduced to combine channel and spatial dimension information for the complex background of GIST images.

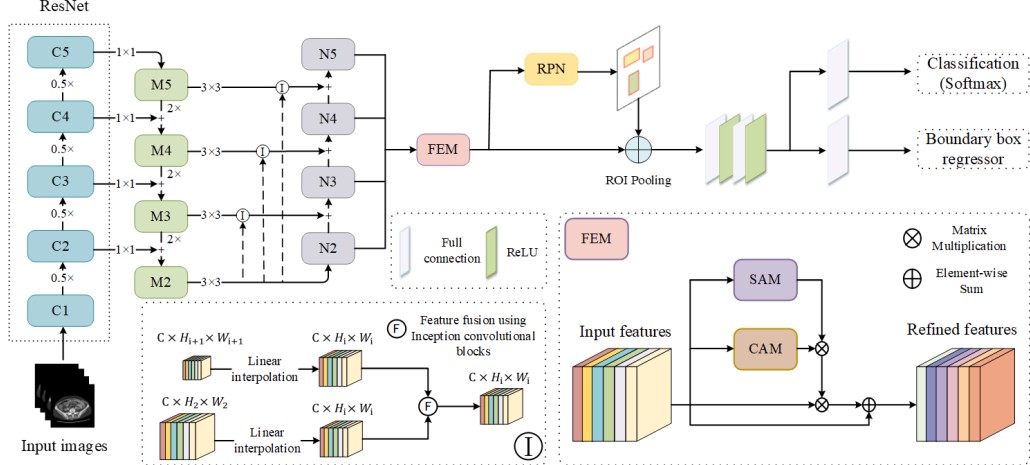

**Figure 2.** The architecture of the Improved Faster R-CNN which consists of several main components: ResNet, the multiscale module, FEM, RPN, and ROI. The input images are processed using ResNet to obtain the feature maps of each layer, and the results are combined using the multiscale module to feed into the FEM. The feature maps enhanced with the FEM are fed into the RPN for anchor proposal, and the ROI then collects the input feature maps and proposals to extract the proposal feature maps into the full connected layer.

One of the challenges of GIST detection is that the object scale varies excessively. Using the single-layer feature map for prediction may affect the accuracy of the result due to the limited information, so the feature maps at different levels should be combined for detection. The traditional feature pyramid network (FPN) [27] can fuse information of the low-level with that of the high-level, but there are still some problems: (1) The transmission path between the low-level features and high-level features is too long, which increases the difficulty of access. (2) Although FPN utilizes the information of different layers, each layer only contains the information of the current layer and higher layers. The lack of location information of lower levels is not conducive to small target detection. In response to the problems in the FPN, we improve it by adding a bottom-up connection based on

the original path. When downsampling the feature map of the $N_i$ layer, $M_2$ and $M_{i+1}$ are bilinearly interpolated to resize to the same scale (the size of the feature map of the $N_i$ layer), and then the fused results are combined with the feature map of the $N_i$ layer to obtain the feature map of the $N_{i+1}$ layer. We choose the Inception [28] convolution block for feature map fusion to solve the problem of excessive computation caused by a large convolution kernel. The improved FPN structure enables the feature map of each layer to contain both the semantic information of the deeper layers and the rich localization information of the first layer, assisting the model in performing better detection.

Another challenge of GIST detection is the difficulty of distinguishing the foreground from the background in CT images. The lesion area shares certain similarities with the surrounding background, and it is hard for the basic model to separate the object, so the FEM is introduced. The feature map obtained through convolution only contains the spatial information in the local receptive field and lacks the connection between each channel. If the information of each channel is only processed globally, the information interaction within the space is missed. Our FEM uses both a channel attention mechanism and a spatial attention mechanism to enhance feature representation, highlight relevant features of the GIST lesion area, and suppress background noise, thus enhancing the feature extraction ability of the network.

We use the channel attention mechanism (CAM) [29] to model the correlation between each channel and obtain the weight of each channel. The process can be written as follows:

$$MLP_1 = Conv_1^{1/r}(Relu(Conv_1^r(P_{\max}(F)))), \tag{1}$$

$$MLP_1 = Conv_1^{1/r}(Relu(Conv_1^r(P_{avg}(F)))), \tag{2}$$

$$CA(X) = \sigma(MLP_1 + MLP_2), \tag{3}$$

where $F$ represents the original feature map, $P_{\max}$ and $P_{avg}$ denote the max pooling and the average pooling, $Conv_1^{1/r}$ denotes that the convolution kernel size is $i \times i$ and the number of channels becomes $1/r$ times of the original, and $\sigma$ is the *Sigmoid*.

The spatial attention mechanism (SAM) [29] is used to model the correlation of the spatial position on the feature map of each channel and calculate its weight. The feature map is calculated as follows:

$$SA(F) = \sigma(Conv_7(P_{\max}(F)\|P_{avg}(F)), \tag{4}$$

where $Conv_7$ denotes that the convolution kernel size is $7 \times 7$, and $\|$ represents merging in the channel dimension.

The FEM refers to BAM [30] and establishes a parallel connection between the CAM and the SAM. Finally, the calculating process can be expressed as follows:

$$\begin{aligned} F'' = FEM(F) &= F + F \times F' \\ &= F + F \times \sigma(expand(CA(F)) \times expand(SA(F))). \end{aligned} \tag{5}$$

### 3.2. Pseudo-Label Selection Strategy

The correctness of pseudo-labels is crucial for subsequent training iterations. If incorrect pseudo-labels are added to the dataset, this will hinder the optimization of model parameters. To this end, we designed a pseudo-label selection strategy based on the dynamic threshold and the IOU constraint, which can effectively screen out pseudo-labels with a higher correct probability and help the model converge.

The method of selecting pseudo-labels by using an unchanging threshold has numerous drawbacks. If the threshold is set too high, the model will filter out the candidate bounding box in the target area and prevent it from being added to the pseudo-label set, leading to a large number of false negative examples in the subsequent training phase. In contrast, if the threshold is set too low, numerous candidate bounding boxes in the nontarget area will be added to the pseudo-label set, thus generating many false-positive

examples in the next round of training. In fact, as training progresses, the network's detecting ability gradually advances, and the validity of the generated pseudo-labels rises. Therefore, the threshold value used to choose the pseudo-label should be dynamic. To avoid incorrect pseudo-labels from influencing model training, we set a high selection threshold at the early stage of training. As training proceeds, we gradually lower this threshold to prevent correct pseudo-labels from being eliminated. Selecting pseudo-labels through the dynamic threshold makes more sense. The value of threshold in the $q^{th}$ round is defined as follows:

$$T_q = \begin{cases} 0.95, q = 1 \\ T_{q-1} - (q-1) \times 0.05, q > 1 \end{cases}. \tag{6}$$

Based on the dynamic threshold, we created a new IOU constraint. The IOU constraint sets the condition for the retention of pseudo-labels. Only when the IOU between the detection results of various transformed images is higher than 0.9 do we regard the bounding box as the pseudo-label. Figure 3 shows the results after applying the IOU constraint. By comparing the original results with the true label, it can be found that the detection box on the right is a false-positive example, which will affect the optimization of the model if it is kept as a pseudo-label. After the IOU constraint is applied, the false-positive bounding box can be successfully excluded, which further ensures the quality of the pseudo labels.

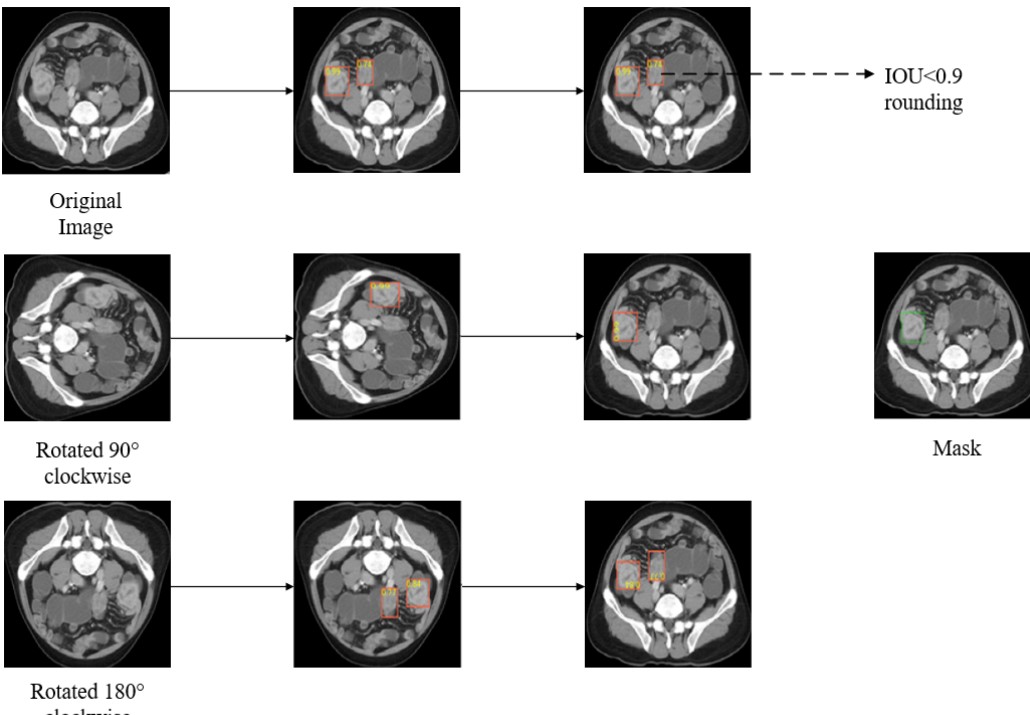

**Figure 3.** The effect of IOU constraint (the orange bounding box in the figure represents the predicted box, and the green bounding box represents the actual labeled box).

By synthesizing the dynamic threshold and the IOU constraint, the selecting criteria of pseudo-labels in the $q^{th}$ round can be expressed as follows:

$$P_i^j = \begin{cases} 1, \ f_i^j > T_q \quad and \quad IOU > 0.9 \\ 0, \ otherwise \end{cases}, \tag{7}$$

where i and j denotes the $j^{th}$ bounding box of the $i^{th}$ image, $f_i^j$ is the confidence of the bounding box, and $P_i^j = 1$ represents that the pseudo-label corresponding to this bounding box is retained, with the opposite being discarded.

*3.3. Self-Training Method*

In this paper, we propose a semi-supervised GIST detection algorithm based on the self-training method, which aims to improve the effectiveness of GIST detection with a small amount of labeled data and a large amount of unlabeled data. The whole procedure of the GIST detection is shown in Algorithm 1.

---

**Algorithm 1** Procedure of Semi-Supervised GIST Detection.

---

**Input:** Labeled data, $L$; Unlabeled data, $U$; Data augmentation strategies $T$ in $\{t_1, t_2, \ldots, t_m\}$
**Output:** Trainable parameters of network, $W$
  1: Initialize hyperparameters: rounds of iteration $Q$, times of data augmentation $K$, threshold $P_1$
  2: Pretrain the model on $L$ to get the initial parameters $W$
  3: **for** $q = 1 : Q$ **do**
  4:     **for** $k = 1 : K$ **do**
  5:         Use $W$ to predict on $t_i(U)$
  6:     **end for**
  7:     $P_q \leftarrow P_q - (q - 1) \times 0.05$
  8:     Filter the results according to (7) to obtain the set of pseudo-labels
  9:     Reassemble to acquire the new training set: $L_n = L \cup R \cup (Mixup(L, R))$
 10:     Retrain the model on $L_n$ to acquire the new parameters $W$
 11: **end for**
 12: **Return** $W$

---

For the labeled data, the labels are the actual bounding boxes, and the confidence is set to 1. We first train with the labeled data to obtain the initial model and then apply different data augmentation strategies to the unlabeled data following the data distillation method proposed by Radosavovic et al. [31].

The data augmentation strategies used in this paper mainly include flip, rotation, and affine transformation. When the flip is chosen as the data augmentation method, the corresponding detection result needs to be flipped as well. For the affine transformation, the set translation range does not exceed 10 pixels, and the position of the bounding box does not vary greatly, so it can remain unchanged. For the rotation operation, the given rotation angle is an integer multiple of 90° or less than 10°. When the angle does not exceed 10°, the position of the bounding box stays unchanged, referring to the affine transformation operation; when the image is rotated 90° clockwise, the coordinates of the corresponding bounding box need to be rotated 90° counterclockwise, and so on for other angles.

The initial model detects the images after data augmentation 1 to k times respectively, and all the results are fused to obtain the pseudo-labels. The pseudo-labels generated by prediction may have some errors. For this reason, we use the dynamic threshold and the IOU constraint to enhance the quality of pseudo-labels. The dynamic threshold refers to a threshold that changes dynamically for the confidence of the pseudo-label, utilizing a higher confidence threshold in the early stages of training and progressively lowering the threshold as training progresses. The IOU constraint is a constraint on the overlap area between the bounding boxes predicted by the initial model on the images after data augmentation 1 to k times. After the transformation, the bounding box is used as a pseudo-label for that image only if it appears on all images with a similar position and size.

After clean pseudo-labels are filtered out through the above-mentioned constraint strategies, Mixup [9] is used to linearly mix the labeled data and the pseudo-labeled data. The new samples acquired after mixing are then used once more for the training, which can substantially enhance the network's generalization capacity.

Mixup is a crucial part of the MixMatch [16] framework, which enables the model to obtain better generalization performance by linearly interpolating pairwise training samples. The traditional Mixup is designed for image classification tasks, where each image is associated with one class label. The generated image $\widetilde{x}$ and its label $\widetilde{y}$ can be defined as follows:

$$\widehat{x} = \lambda x + (1-\lambda)x', \tag{8}$$
$$\widehat{y} = \lambda y + (1-\lambda)y', \tag{9}$$

where $x$ and $x'$ denote two different images, $y$ and $y'$, respectively, denote their probability of the corresponding class, $\lambda \in [0,1]$.

Since the data used in this paper are annotated with the bounding box of the lesion, we opted for image-level Mixup rather than classification Mixup. The generated label $\widehat{x}_i$ and its confidence $\widehat{y}_i$ in image-level Mixup can be calculated as follows:

$$\widetilde{\lambda} = \max(\lambda, 1-\lambda), \tag{10}$$
$$\widehat{x}_i = \widetilde{\lambda}x_i + \left(1-\widetilde{\lambda}\right)x'_i, \forall i, \tag{11}$$
$$\widehat{y}_i = \widetilde{\lambda}y_i + \left(1-\widetilde{\lambda}\right)y'_i, \forall i, \tag{12}$$

where $\widehat{x}_i$ denotes the $i^{th}$ label on the generated image $\widehat{x}$, $x$ and $x'$ denote bounding boxes on two different images, and $y$ and $y'$, respectively, represent their confidence, $\lambda \in [0,1]$.

## 4. Results

We first conducted ablation studies to verify the effectiveness of the proposed module and strategies. Furthermore, we designed experiments to demonstrate the superiority of the Improved Faster R-CNN and the self-training method. Detailed information on the configuration and results is presented in the following subsections.

### 4.1. Datasets and Experimental Settings

Datasets used for the fully supervised method (Improved Faster R-CNN) were the following: The datasets used in the experiment were the CT images (a series of DICOM format files obtained by doctors using related equipment to scan the patient's abdomen) of GIST patients provided by the hospital, including labeled images of 213 patients and unlabeled images of 10 nonpatients. Each patient had between 50 and 80 slices, of which only 3 to 10 slices contained GIST. The slices with lesions labeled by qualified medical professionals were used as the datasets in this work. We used pydicom to convert DICOM format files to png format images and finally obtained a total of 3735 images with GIST annotated by doctors, comprising 526 ones with the small object, 2212 ones with the medium object, and 997 ones with the large object. Of the labeled images, 70% were used for training, and the remaining 30% of labeled images were added to the test set together with 600 slices without lesions of 10 nonpatients. Table 1 displays the makeup of the training set and test set for the GIST detection experiment.

Experimental settings in the fully supervised method (Improved Faster R-CNN) were the following: The operating system was an Ubuntu 18.04, and the hardware environment was an Intel(R) Core(TM) i9-10980XE CPU@3.00 GHz and two TITAN RTX 24 G graphics cards. The programming language used was python3.7, and the framework was the PyTorch-based mmdetection [32]. The backbone of the network was ResNet50, the number of training epochs was 24, the batch size was 8, the optimizer was stochastic gradient descent (SGD), the momentum was 0.9, and the initial learning rate was 0.01, which decays at epochs 16 and 22.

**Table 1.** Composition of the training set and test set.

| Dataset | Object | Number of Slices |
|---|---|---|
| Training set | Small-scale | 413 |
| | Medium-scale | 1412 |
| | Large-scale | 789 |
| Test set | Small-scale | 113 |
| | Medium-scale | 800 |
| | Large-scale | 208 |
| | Nonlesion | 600 |

Datasets used for the semi-supervised method (Improved Faster R-CNN) were the following: The semi-supervised method experiment requires a greater number of unlabeled samples, so the datasets used for the fully supervised method were divided, using 3%, 5%, 10%, and 20% of the data as labeled samples, and the remaining data were unlabeled after the labels were removed to form an unlabeled dataset together with the unlabeled data of 54 patients. The specific division of the train set used for the semi-supervised method is shown in Table 2. The test set was consistent with that used in the fully supervised method experiment.

**Table 2.** Composition of the training set.

| Proportion of Retained Labels | Labeled Data | Unlabeled Data |
|---|---|---|
| 3% | 90 | 6180 |
| 5% | 150 | 6120 |
| 10% | 300 | 5970 |
| 20% | 600 | 5670 |

The experimental settings in the semi-supervised method were as follows: The hardware environment was identical to that in the fully supervised method experiment, and the Improved Faster R-CNN was used as the object detector for the experiments. The batch size was set to 8, the optimizer was stochastic gradient descent (SGD), the initial learning rate was 0.01, the momentum was 0.9, the rounds of iterations were 5, and the times of data augmentation were 4. When training the network with only labeled data, the number of epochs was set to 30 to obtain a better-performing initial model and higher-quality pseudo-labels, with the learning rate decaying at epochs 18 and 26. For better comparison with the experimental results of the fully supervised method, the experimental settings when unlabeled data were used for training were the same as those in the fully supervised learning experiments. The number of epochs was set to 24, and the learning rate decayed at epochs 16 and 22.

The variables used in the self-training method experiments included Q and the threshold. Q indicates the total number of training rounds after adding pseudo-labels, and a higher value of Q indicates a higher proportion of pseudo-labels in the dataset used for training. The pseudo-label selection threshold is one of the criteria used to select pseudo-labels during the training process. If the score of the prediction result is lower than the threshold, it cannot be used as a pseudo-label. The threshold value reflects the correctness of the pseudo-label, and the larger the threshold value is set, the higher the correctness of the pseudo-label.

### 4.2. Ablation Study on Improved Faster R-CNN

In this section, we describe the designed experiments conducted to demonstrate the effectiveness and superiority of the multiscale module and FEM introduced in the Improved Faster R-CNN. Table 3 lists the experimental results.

**Table 3.** Investigation of different modules introduced to the Faster R-CNN.

| Model | FPN | PAFPN | Ours_FPN | FEM | $AP_s$ | $AP_m$ | $AP_l$ | AP |
|---|---|---|---|---|---|---|---|---|
| | | | | | 0.329 | 0.677 | 0.796 | 0.662 |
| | ✓ | | | | 0.395 | 0.692 | 0.806 | 0.680 |
| | | ✓ | | | 0.413 | 0.699 | 0.815 | 0.701 |
| Faster R-CNN | | | ✓ | | 0.431 | 0.702 | 0.827 | 0.735 |
| | | | | ✓ | 0.342 | 0.671 | 0.799 | 0.667 |
| | ✓ | | | ✓ | 0.401 | 0.695 | 0.811 | 0.682 |
| | | ✓ | | ✓ | 0.415 | **0.711** | 0.814 | 0.705 |
| | | | ✓ | ✓ | **0.435** | 0.704 | **0.829** | **0.739** |

The bold indicates the best result.

We respectively compare the effects of the Faster R-CNN without the multiscale module, with the initial FPN [27] module, with the PAFPN [33] module, and with the FPN module proposed in this paper, and the experimental results are shown in Table 3 and Figure 4. The data in the table show that the proposed FPN structure increases the AP of the entire item by 0.073, and the AP at all scales is improved by the improved FPN structure, with the most significant improvement for small objects. The suggested multiscale approach surpasses other methods in the AP of both the overall objects and each scale object, proving that the improved FPN is more effective for the task at hand.

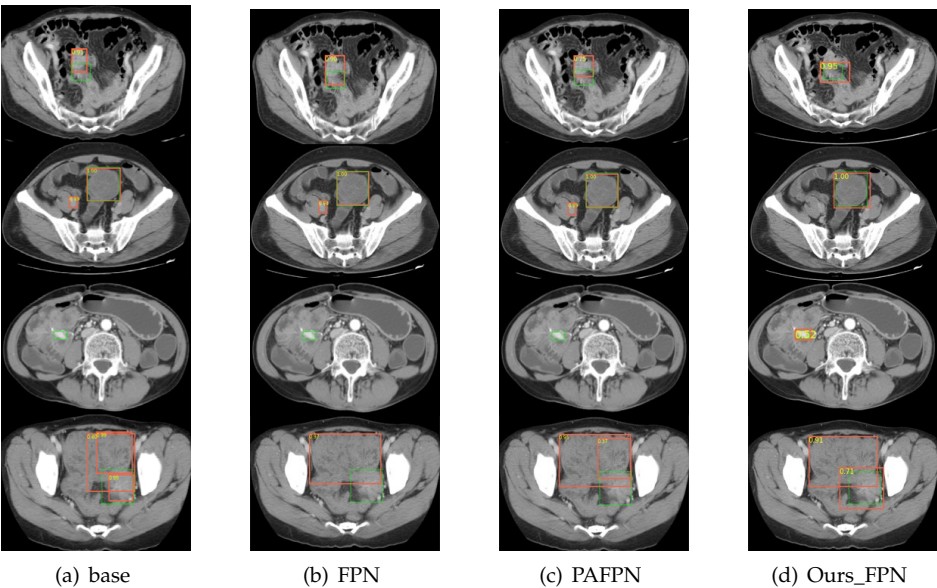

|  (a) base | (b) FPN | (c) PAFPN | (d) Ours_FPN |

**Figure 4.** Visual comparison of different feature pyramid networks (the orange bounding box in the figure represents the predicted box, and the green bounding box represents the actual labeled box).

As shown in Table 3 and Figure 5, we compared the test results using only the FEM, using the FPN module together with the FEM, using the PAFPN module together with the FEM, and using our FPN together with the FEM to verify the effectiveness of the FEM. As evident from the results in the table, the AP is slightly improved when using only the FEM without the multiscale module. Using FPN as the multiscale module, the FEM yields an AP improvement of 0.002, using PAFPN 0.004, and using the proposed multiscale module 0.04. Overall, the AP goes from 0.662 to 0.739 with the addition of the multiscale module and the FEM. Although the effect of the FEM on this task is not as significant as that of the multiscale module, a series of comparative experiments also proved that this module can improve the performance of the network.

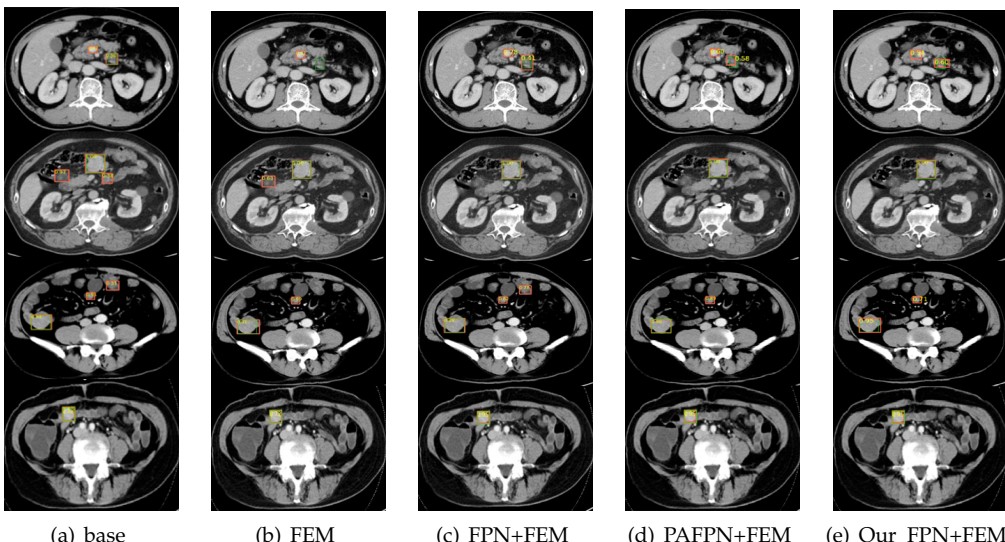

(a) base     (b) FEM     (c) FPN+FEM     (d) PAFPN+FEM     (e) Our_FPN+FEM

**Figure 5.** Visual comparison of the FEM combined with the different feature pyramid networks (the orange bounding box in the figure represents the predicted box, and the green bounding box represents the actual labeled box).

### 4.3. Ablatation Study on the Semi-Supervised Method

In this section, we detail the ablation experiments of the SSL method to demonstrate that the strategies we introduced to the self-training method are effective in improving the detection results related to the task defined in this paper. In addition, we explored the impact of different settings on the model performance.

To verify the effectiveness of the self-training method used in this paper, we first iteratively optimized the initial model using the original self-training method. We usd the control variable method to test the two initialization hyperparameters, the number of training rounds, and the threshold for pseudo-label selection. The experimental results are shown in Table 4, where Q represents the total number of iterations. When Q = 0, only the labeled data are used for training, and when Q > 0, pseudo-labels are gradually added to the train set.

Table 4 shows that as Q rises, the detection accuracy declines. This phenomenon indicates that in the GIST detection task, accuracy improvement cannot be achieved by using the initial self-training method to create pseudo-labels. With the threshold held constant, we observe that as the round of iteration increases, the accuracy of the training set with 3% labeled data decreases faster than that with 20% labeled data. Because the initial model trained with 3% labeled data is less accurate than that with 20% labeled data, a greater number of false pseudo-labels are generated as the number of rounds increases. In the same training set, we can find that the accuracy decreases faster when the pseudo-label selection threshold is 0.5. Using a lower threshold leads to many bounding boxes in nonlesion regions being added to the pseudo-label set, thus generating many false-positive samples in the subsequent round. When the pseudo-label selection threshold is 0.9, the accuracy of the model improves briefly, but as the number of iteration rounds increases, the model tends to overfit the high-confidence data, resulting in a decrease in accuracy. Therefore, if the accuracy of the initial model is too low or the threshold value is not appropriate, there will be too many noisy labels. This result shows that using the original self-training strategy reduces, rather than improves, the detection accuracy due to noisy pseudo labels.

**Table 4.** Investigation of the proportions of labelled data, the thresholds, and the rounds.

| Training Set | Threshold | Q = 0 | Q = 1 | Q = 2 | Q = 3 |
|---|---|---|---|---|---|
| 3% Labeled data | 0.5 | **0.200** | 0.154 | 0.126 | 0.115 |
| | 0.7 | **0.200** | 0.163 | 0.137 | 0.125 |
| | 0.9 | 0.200 | 0.202 | **0.205** | 0.203 |
| 5% Labeled data | 0.5 | **0.312** | 0.281 | 0.241 | 0.239 |
| | 0.7 | **0.312** | 0.288 | 0.246 | 0.244 |
| | 0.9 | 0.312 | **0.314** | 0.313 | 0.312 |
| 10% Labeled data | 0.5 | **0.501** | 0.481 | 0.472 | 0.469 |
| | 0.7 | **0.501** | 0.482 | 0.475 | 0.471 |
| | 0.9 | 0.501 | 0.503 | **0.505** | 0.504 |
| 20% Labeled data | 0.5 | **0.637** | 0.615 | 0.608 | 0.604 |
| | 0.7 | **0.637** | 0.621 | 0.614 | 0.611 |
| | 0.9 | 0.637 | **0.638** | 0.636 | 0.636 |

The bold indicates the best result.

Table 5 shows the the experimental findings on the 10% annotated dataset without data augmentation, with data augmentation using horizontal flip, vertical flip, random rotation, random noise, and affine transformation to confirm the impact of various data augmentation approaches. The use of data augmentation techniques other than random noise can increase the accuracy of detection. One of the four data augmentation techniques (horizontal flip, vertical flip, rotation, and affine transformation) was chosen at random in the experiment. There is some variation in the training outcomes because the data augmentation method was chosen at random. As a result, the findings of the subsequent experiments involving data augmentation were averaged after four rounds of training.

**Table 5.** Influence of using different data augmentation methods on the 10% labeled dataset.

| Data Augmentation | $AP_{0.5}$ | $AP_{0.75}$ | AP |
|---|---|---|---|
| Base | 0.793 | 0.580 | 0.501 |
| Horizontal flip | 0.801 | 0.585 | 0.506 |
| Verical flip | 0.795 | <u>0.578</u> | 0.504 |
| Random rotation | 0.794 | 0.581 | 0.503 |
| Affine transformation | 0.794 | 0.582 | 0.502 |
| Random noise | 0.801 | <u>0.576</u> | <u>0.491</u> |

The underline indicates the worse results after using data augmentation.

To demonstrate the effectiveness of the dynamic threshold, the IOU constraint, and the Mixup used in the self-training method, we present ablation studies on the 10% labeled data. Table 6 shows that after applying the dynamic threshold and the IOU constraint, the results are improved with iteration. In contrast, as the training iterates, the accuracy of the model using the original self-training method declines. We can also observe that employing Mixup results in a modest rise in AP, proving that Mixup enhances the network's generalization ability and improves the model's performance.

**Table 6.** Investigation of using different strategies in the self-training method.

| Model | Dynamic Threshold | IOU Constraint | Mixup | Q = 0 | Q = 3 | Q = 5 |
|---|---|---|---|---|---|---|
| | ✓ | | | 0.501 | 0.515 | 0.518 |
| Improved | | ✓ | | 0.501 | 0.517 | 0.518 |
| Faster R-CNN | ✓ | ✓ | | 0.501 | 0.519 | 0.521 |
| | ✓ | ✓ | ✓ | 0.501 | **0.521** | **0.524** |

The bold indicates the best result.

## 4.4. Comparison Experiments and Analysis

In this section, we describe the series of comparative experiments conducted to prove the superiority of the Improved Faster R-CNN and the proposed self-training method.

As shown in Table 7 and Figure 6, the Improved Faster R-CNN outperformed the other mainstream object detection algorithms, including the one-stage object detector, two-stage object detector, and anchor-free object detector, which demonstrates that our network has superiority by virtue of the multiscale module and the FEM.

**Table 7.** Comparison with the state-of-the-art object detection algorithms.

| Model | $AP_{0.5}$ | $AP_{0.75}$ | AP |
|---|---|---|---|
| Faster R-CNN [1] | 0.923 | 0.771 | 0.662 |
| Mask R-CNN [34] | 0.933 | 0.802 | 0.702 |
| Cascade R-CNN [2] | 0.914 | 0.798 | 0.703 |
| YOLOv3 [3] | 0.933 | 0.474 | 0.484 |
| RetinaNet [4] | 0.922 | 0.783 | 0.673 |
| FCOS [35] | 0.860 | 0.543 | 0.516 |
| Improved Faster R-CNN | **0.961** | **0.802** | **0.739** |

The bold indicates the best result.

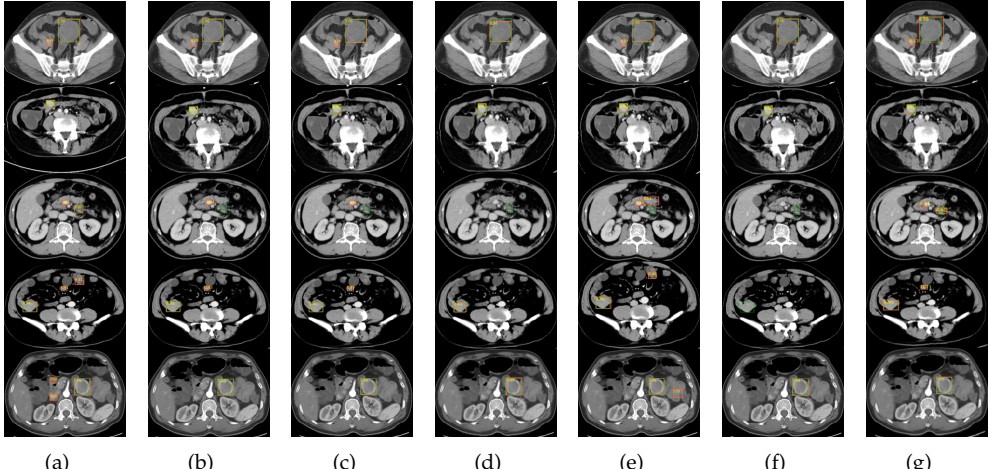

|  (a)  |  (b)  |  (c)  |  (d)  |  (e)  |  (f)  |  (g)  |

**Figure 6.** Visual comparison of the different object detection algorithms including (**a**) Faster R-CNN, (**b**) Mask R-CNN, (**c**) Cascade R-CNN, (**d**) YOLOv3, (**e**) RetinaNet, (**f**) FCOS, and (**g**) Improved Faster R-CNN. The orange bounding box in the figure represents the predicted box, and the green bounding box represents the actual labeled box.

In Table 8, we present the improvement in accuracy when using our semi-supervised method compared to CSD [23], STAC [24], and Instant-teaching [36] on all datasets. It demonstrates that the network has higher robustness for all datasets since the dynamic threshold and the IOU constraint guarantee accurate pseudo-labels.

**Table 8.** Comparison with state-of-the-art semi-supervised methods.

| Dataset | Method | $AP_{0.5}$ | $AP_{0.75}$ | AP |
|---|---|---|---|---|
| 3% labeled data | CSD | 0.283 | 0.227 | 0.198 |
| | STAC | 0.291 | 0.231 | 0.191 |
| | Instant-teaching | 0.288 | 0.246 | 0.197 |
| | Ours | **0.301** | **0.257** | **0.204** |
| 5% labeled data | CSD | 0.574 | 0.346 | 0.351 |
| | STAC | **0.597** | 0.319 | 0.348 |
| | Instant-teaching | 0.574 | 0.300 | 0.341 |
| | Ours | 0.587 | **0.412** | **0.355** |
| 10% labeled data | CSD | 0.715 | 0.511 | 0.499 |
| | STAC | 0.759 | 0.587 | 0.508 |
| | Instant-teaching | 0.781 | 0.604 | 0.504 |
| | Ours | **0.835** | **0.611** | **0.524** |
| 20% labeled data | CSD | 0.891 | 0.721 | 0.639 |
| | STAC | 0.901 | 0.701 | 0.638 |
| | Instant-teaching | 0.913 | 0.728 | 0.648 |
| | Ours | **0.932** | **0.731** | **0.659** |

The bold indicates the best result.

Finally, we compared the semi-supervised method with the fully supervised Improved Faster R-CNN on the fully labeled dataset, using all the labeled data for initial training and adding the unlabeled CT images of only 54 patients for the pseudo-label generation. The experimental results are shown in Table 9 and Figure 7. The comparison reveals that the semi-supervised method can use the information in the unlabeled data to produce detection results that are marginally better than those of the fully supervised method after using unlabeled data. It proves that the addition of unlabeled data has no impact on the model's performance. However, the improvement in the model performance is not substantial because the unlabeled data used in this comparison experiment were insufficient and only contained images of 54 patients.

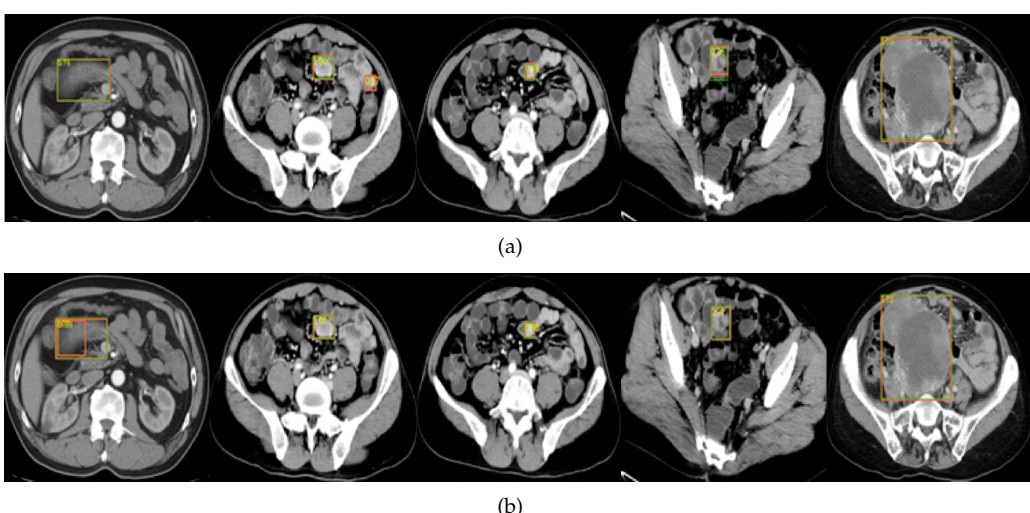

(a)

(b)

**Figure 7.** Visual comparison of (**a**) the fully supervised method and (**b**) semi-supervised method. The orange bounding box in the figure represents the predicted box, and the green bounding box represents the actual labeled box.

**Table 9.** Comparison with the fully supervised Improved Faster R-CNN.

| Method | AP$_{0.5}$ | AP$_{0.75}$ | AP |
|---|---|---|---|
| Fully supervised Improved Faster R-CNN | 0.959 | **0.799** | 0.739 |
| Semi-supervised Improved Faster R-CNN | **0.960** | 0.797 | **0.740** |

The bold indicates the best result.

## 5. Conclusions

To address the problems of large object scale variation, confusing background, and the challenge of obtaining labeled data in the detection of GIST, we propose a semi-supervised object detection method using self-training in this study. The method uses only a small amount of labeled data supplemented by a sizable amount of unlabeled data and fully exploits the information contained in the unlabeled data. Through comparison with existing methods and ablation studies of each module, the feasibility of the proposed method was proven, and the detection accuracy of the model increased without extra labeling costs.

Although the improved scheme for GIST detection in this paper has achieved good results, there are still some limitations that should be noted. In the semi-supervised learning method, the best prediction results are used as pseudo-labels for subsequent training. The challenging samples—those that the model has not yet learned well—are not used effectively. Therefore, we can try to combine semi-supervised learning and active learning to find challenging samples by active learning and then manually label the challenging samples for training.

**Author Contributions:** Conceptualization, Q.Y.; methodology, Q.Y.; software, Y.J.; validation, Z.C.; formal analysis, Z.C.; investigation, Y.J.; resources, F.M.; data curation, H.S.; writing—original draft preparation, Z.C.; writing—review and editing, X.G.; visualization, H.S.; supervision, F.X.; project administration, Q.Y.; funding acquisition, G.G. All authors have read and agreed to the published version of the manuscript.

**Funding:** This work was partially supported by the National Natural Science Foundation of China, Nos. 61973250, 62073218, 61973249, 61902316, 61902313, 62002271, 82150301, 62133012, 62273232, 62273231. Young science and technology nova of Shaanxi Province (2022KJXX-73) and the Fundamental Research Funds for the Central Universities under grant No. XJS210310. The Shaanxi Provincial Department of Education Serving Local Scientific Research (19JC038), the Key Research and Development Program of Shaanxi (2021GY-077), the Young Science and Technology Nova of Shaanxi Province (2022KJXX-73), the Shanghai Municipal Health Commission (202140512), and the Shanghai Stroke Association (SSA-2020-023). National Key R and D program of China (2022YFB4300700), the Key R and D programs of Shaanxi Province (2021ZDLGY02-06), Qin Chuangyuan project (2021QCYRC4-49), Qinchuangyuan Scientist+Engineer (2022KXJ-169), National Defense Science and Technology Key Laboratory Fund Project (6142101210202).

**Institutional Review Board Statement:** Not applicable.

**Informed Consent Statement:** Informed consent was obtained from all subjects involved in the study.

**Data Availability Statement:** The data that support the findings of this study are available on request from the corresponding author. The data are not publicly available due to privacy or ethical restrictions.

**Conflicts of Interest:** The authors declare no potential conflict of interest.

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
