# Peer review of "Semi-Supervised Gastrointestinal Stromal Tumor Detection via Self-Training"

_electronics, doi:10.3390/electronics12040904_

Round 1

Reviewer 1 Report

The manuscript aims to proffer a semi-supervised technique for identifying GIST utilizing self-training, which necessitates a paucity of labeled data and a plethora of unlabelled data. The authors purport that this methodology generates a heightened insight without incurring additional expenses associated with labeling. The authors have executed an impeccable job in terms of experimental design, outlining the obstacles, and articulating the solution in a cogent manner. I discourse that the manuscript is eminently suitable for publication in the journal "Electronics" provided that the minor issues highlighted are addressed.

Minor points:

1.      The abstract should have one or two sentences about the significance of detecting GIST.

2.      The authors need to discuss the current GIST detection techniques.

3.      Section 2 needs a better literature survey, specifically the impact of artificial intelligence on diagnosing GIST.

4.      Are authors claiming that the faster R-CNN they have proposed is novel and has not been done before by others?

5.      Authors need to have a section with limitations and constraints of their study and possible way out.

I would say that the authors in question are skilled in experimental design and data analysis. They have a thorough understanding of the methodology used in their experiment and have been able to execute it effectively. Furthermore, they have been able to critically evaluate the results of their experiment and draw meaningful conclusions from their data. Overall, they have demonstrated good expertise in their field and contributed valuable insights through their research.

Author Response

Point 1: The abstract should have one or two sentences about the significance of detecting GIST.

Response 1: We sincerely appreciate your valuable comment. We have added a note about the significance of GIST detection in the ABSTRACT part of the revised manuscript (Paragraph 1, Page 1).

Point 2: The authors need to discuss the current GIST detection techniques.

Response 2: Thanks for the helpful suggestion. We have discussed the current GIST detection techniques in the INTRODUCTION part according to the comment (Paragraph 2, Page 1; Paragraph 3, Page 2).

Point 3: Section 2 needs a better literature survey, specifically the impact of artificial intelligence on diagnosing GIST.

Response 3: As suggested by the reviewer, we have added a brief introduction to elucidate the impact of artificial intelligence on the diagnosis of GIST in the RELATED WORK part (Paragraph 1, Page 3). Also, we cited Fei’s original work referenced as [35].

Point 4: Are authors claiming that the faster R-CNN they have proposed is novel and has not been done before by others?

Response 4: Thank you for the critical comment. We proposed a novel neural network based on Faster R-CNN, which is a novel framework with the multiscale module and the feature enhancement module. These modules are designed for the characteristics of GIST, resulting in the improvement of the accuracy of GIST detection.

Point 5: Authors need to have a section with limitations and constraints of their study and possible way out.

Response 5: Thanks for your suggestion. We have added a section with limitations and constraints of our study and possible way out in the CONCLUSION part (Paragraph 3, Page 16).

Reviewer 2 Report

Semi-supervised Gastrointestinal Stromal Tumor Detection via 4 self-training The concepts of the paper have sound, and the authors are appreciated for choosing this innovative topic. However, the authors must resolve the following issues addressed in this research article. 1. The novelty of the paper needs to be framed with respect to the recent literature 2. The model is described in all its details, but I don't see the link to the literature. 3. Section 4 should be called results. An explanation of the results occurs. However, some aspects need to be strengthened: a. Identification of critical variables; b. The need for values to vary over time and whether they affect the final results; c. Trade-offs between some variables 4. State the limitations in the conclusions and applications that can be obtained from the results of this work

Author Response

Point 1: The novelty of the paper needs to be framed with respect to the recent literature.

Response 1: Thanks for your suggestion. We have added the discussion of the recent literature in the INTRODUCTION part (Paragraph 2, Page 1; Paragraph 3, Page 2).

Point 2: The model is described in all its details, but I don't see the link to the literature.

Response 2: We sincerely appreciate the valuable comment. We have described the steps of each iteration in the self-training method (Figure Legend, Figure 1) and the input and output of each module in the network (Figure Legend, Figure 2).

Point 3: Section 4 should be called results.

Response 3: We have re-named Section 4 as suggested by the reviewer according to the comment.

Point 4: some aspects need to be strengthened: a. Identification of critical variables;

Response 4: As suggested by the reviewer, we have added the identification of critical variables in the RESULTS part (Paragraph 3, Page 10).

Point 5: some aspects need to be strengthened: b. The need for values to vary over time and whether they affect the final results;

Response 5: We have explained the reasons and effects of changing the number of epochs in the semi-supervised method experiments in the RESULTS part according to the comment (Paragraph 2, Page 10).

Point 6: some aspects need to be strengthened: c. Trade-offs between some variables.

Response 6: Thanks for your suggestion. However, there are no constraints or mutually exclusive relationships between the variables in our experiments.

Point 7: State the limitations in the conclusions and applications that can be obtained from the results of this work.

Response 7: We have added a section with limitations and constraints of our study and possible ways out in the CONCLUSION part according to the comment (Paragraph 3, Page 16).
